# Online Segment Any 3D Thing as Instance Tracking

**Hanshi Wang**[1,2,3,5][*] **Zijian Cai**[3]**, Jin Gao**[1,2,5][†]**,Yiwei Zhang**[1,2,5]**,**
**Weiming Hu**[1,2,5,6]**, Ke Wang**[7]**, Zhipeng Zhang**[3,4][†]
[1]State Key Laboratory of Multimodal Artificial Intelligence Systems (MAIS), CASIA
[2]School of Artificial Intelligence, University of Chinese Academy of Sciences
[3]AutoLab, School of Artificial Intelligence, Shanghai Jiao Tong University [4]Anyverse Intelligence
[5]Beijing Key Laboratory of Super Intelligent Security of Multi-Modal Information
[6]School of Information Science and Technology, ShanghaiTech University [7]KargoBot
{hanshi.wang.cv, zhipeng.zhang.cv}@outlook.com; jin.gao@nlpr.ia.ac.cn

## Abstract

Online, real-time, and fine-grained 3D segmentation constitutes a fundamental capability for embodied intelligent agents to perceive and comprehend their operational environments. Recent advancements employ predefined object queries to aggregate semantic information from Vision Foundation Models (VFMs) outputs that are lifted into 3D point clouds, facilitating spatial information propagation through inter-query interactions. Nevertheless, perception, whether human or robotic, is an inherently dynamic process, rendering temporal understanding a critical yet overlooked dimension within these prevailing query-based pipelines. This deficiency in temporal reasoning can exacerbate issues such as the over-segmentation commonly produced by VFMs, necessitating more handcrafted post-processing. Therefore, to further unlock the temporal environmental perception capabilities of embodied agents, our work reconceptualizes online 3D segmentation as an instance tracking problem (AutoSeg3D). Our core strategy involves utilizing object queries for temporal information propagation, where long-term instance association promotes the coherence of features and object identities, while short-term instance update enriches instant observations. Given that viewpoint variations in embodied robotics often lead to partial object visibility across frames, this mechanism aids the model in developing a holistic object understanding beyond incomplete instantaneous views. Furthermore, we introduce spatial consistency learning to mitigate the fragmentation problem inherent in VFMs, yielding more comprehensive instance information for enhancing the efficacy of both long-term and short-term temporal learning. The temporal information exchange and consistency learning facilitated by these sparse object queries not only enhance spatial comprehension but also circumvent the computational burden associated with dense temporal point cloud interactions. Our method establishes a new state-of-the-art, surpassing ESAM by 2.8 AP on ScanNet200 and delivering consistent gains on ScanNet, SceneNN, and 3RScan datasets, corroborating that identity-aware temporal reasoning is a crucial, previously underemphasized component for robust 3D segmentation in real-time embodied intelligence. Code is at `https://github.com/AutoLab-SAI-SJTU/AutoSeg3D`.

## 1 Introduction

The ability to perform online, real-time, and fine-grained 3D instance segmentation is a cornerstone for embodied intelligent agents to perceive and comprehend their operational environments. Autonomous robots and embodied assistants increasingly depend on such systems for exploring and interacting

---

[*]This work was completed during Hanshi's remote internship at SJTU and co-mentored by Prof. Zhipeng Zhang. [†]Corresponding author.

39th Conference on Neural Information Processing Systems (NeurIPS 2025).

with complex scenes. Early approaches predominantly adopts an offline paradigm, which involved accumulating complete point clouds prior to processing, thereby incurring prohibitive latency and memory costs. In pursuit of faster and online perception capabilities, recent research has begun to explore paradigms assisted by Vision Foundation Models (VFMs) such as SAM [1].

Current online VFM-assisted models are engineered to process streaming inputs by initially predicting segmentation results with VFMs and subsequently lifting the generated masks and recorded depth to superpoint representations. However, these pipelines simply concatenate global point features across scans and omit instance level temporal modeling, which worsens fragmentation and over segmentation by VFMs. Post hoc non-maximum suppression only partially corrects these errors and introduces the concurrent loss of valid information not as expected.

Seeking to address these limitations, we draw inspiration from established methodologies for maintaining temporal coherence in online perception. Classical multi-object tracking (MOT) methods, for instance, achieve consistent identity assignment by exploiting spatial continuity and appearance affinities to link detections across frames [2, 3]. Similarly, video instance segmentation frameworks like VisTR [4] and 3D detection models such as Sparse4D [5] employ query-based memory banks to propagate and update object features over time, enabling each instance to maintain a persistent representation robust to occlusion and partial views. The core design principle underpinning these diverse approaches is the explicit maintenance and evolution of instance-specific representations across temporal sequences. Inspired by this paradigm, we recast online 3D instance segmentation as an instance-tracking task. By integrating object-level temporal priors directly into the segmentation pipeline, our approach aims to concurrently rectify over-segmentation errors and enforce identity consistency, thereby substantially enhancing overall segmentation performance and robustness.

More specifically, we introduce a novel, tracking-centric pipeline that directly addresses the two core limitations of VFM-based methods. Our framework decomposes into three lightweight and synergistic modules. First, the Long-Term Memory (LTM) maintains a bounded track bank and employs Hungarian assignment based on confidence-gated affinity matrix to recover identities after prolonged occlusions with constant overhead. Second, the Short-Term Memory (STM) refines instance embeddings via distance-aware cross-frame attention to inject immediate temporal context while filtering out background noise. Third, Spatial Consistency Learning (SCL) merges high-affinity mask fragments at inference by jointly reasoning over 2D appearance and 3D geometry, while concurrently employing one-to-many fragment supervision during training to mitigate over-segmentation and generate coherent, high-fidelity queries for LTM and STM. Together, these components form a cohesive, real-time 3D instance segmentation system that enforces consistent object identities across frames, injects immediate temporal context while filtering out background noise, and merges high-affinity fragments to directly counteract VFM over-segmentation. By integrating these modules, our framework preserves real-time throughput while delivering a 2.8 AP gain over recent ESAM [6] on ScanNet200 [7]. Extensive evaluations on both ScanNet200 and ScanNet [8], as well as zero-shot assessments on SceneNN [9] and 3RScan [10] demonstrate consistent performance gains.

In summary, our contributions are as follows: **1)** We recast online 3D instance segmentation as a continuous instance tracking problem by treating each VFM-derived mask as a track query within a unified framework. **2)** We propose a lightweight architecture with three synergistic modules where LTM propagates identities across frames to ensure continuity, STM injects short-term temporal context while filtering background noise, and SCL merges overlapping fragments to counteract over-segmentation and enrich instance embeddings. **3)** Our framework achieves new state-of-the-art results on ScanNet200, ScanNet, SceneNN, and 3RScan while sustaining real-time throughput, and ablation studies verify the contribution of each component.

## 2   Related Work

**VFM-assisted 3D Scene Segmentation.** Vision foundation models (VFMs) have emerged as a promising cornerstone for 3D scene understanding in embodied intelligence, especially in the construction and reasoning of 3D spatial information [11, 12, 13, 14, 1, 15, 16, 17, 18]. Large-scale pretrained VFMs such as SAM [1] and CLIP [12] exhibit powerful open-vocabulary segmentation and semantic alignment capabilities, which have been extensively leveraged in downstream 3D perception pipelines. SAM3D [19] first predicts 2D instance masks with SAM and then lifts them to 3D via depth and camera parameters, followed by geometric merging. CLIP2Scene [20] distills multimodal knowl-

edge from CLIP into a 3D backbone through semantic and spatio-temporal consistency regularization, enabling label-efficient scene parsing. OpenMask3D [21] combines CLIP-extracted visual features with SAM-refined masks to generate discriminative per-instance embeddings for open-vocabulary 3D instance segmentation. Despite these advances, several studies have highlighted that the 2D masks produced by VFMs are often over-segmented. SAI3D [22], for example, decomposes the reconstructed mesh into 3D primitives, assigns semantic scores to 2D masks via Semantic-SAM [15], and aggregates the primitives through a graph-based region-growing algorithm. Nevertheless, existing approaches still rely on heuristic post-hoc fusion of projected 3D masks, which often proves brittle in cluttered or dynamically changing robotic environments. In this work, we propose a learnable fusion module that jointly reasons over over-segmentation hypotheses in both 2D and 3D spaces. By optimizing fusion in an end-to-end manner, our method mitigates the impact of erroneous 2D masks and delivers more robust and scalable 3D scene understanding.

**Online 3D Scene Perception.** Driven by the rapid advancements in autonomous driving and embodied AI, robotic tasks are increasingly demanding higher levels of 3D scene understanding. In these scenarios, the ability to process information in real time, adapt to diverse conditions, and achieve perception is crucial. However, most of the common instance segmentation methods [23, 24, 25, 26, 27, 28, 29, 21, 30, 31, 32, 33] are offline. They can handle large-scale datasets but are highly dependent on the quality of preprocessing and data augmentation, which makes it difficult to apply them to complex and ever-changing robotic environments. Recently, online 3D scene perceptions [34, 35, 36, 37, 38, 39] have attracted increasing attention. INS-Conv [38] proposes an incremental sparse convolutional network for online 3D segmentation, which achieves efficient and accurate inference by processing only the residuals between consecutive frames and incorporating an uncertainty term to adaptively select which residuals to update. MemAda [40] proposes an adapter-based model that equips mainstream offline frameworks with the competence to perform online scene perception, enabling them to process real-time RGB-D sequences efficiently. Building on this foundation, ESAM [6] further advances the field by achieving online scene segmentation and designing a dual-layer decoder along with auxiliary tasks to facilitate the merging of 3D masks. While prevailing methods fuse dense features (*e.g.,* raw point clouds) temporally, they often lack the semantic context crucial for instance-level tasks. We address this by recasting online segmentation as instance tracking, which allows us to propagate semantically rich instance information across frames. This focus on semantic consistency through time yields significantly more precise instance segmentation results, while also being computationally efficient.

## 3 Method

### 3.1 Overall Architecture

Fig. 3.1 illustrates our tracking-centric online 3D segmentation framework. The design draws inspiration from the brain's complementary learning systems [41, 42, 43, 44, 45]. Specifically, the hippocampus rapidly forms episodic memories, allowing quick adaptation to novel contexts and interaction with recent experience, whereas the neocortex consolidates these transient traces into durable representations through slow, cumulative learning, producing a stable store of knowledge. This dual mechanism not only enhances adaptability but also ensures the coherence and persistence of memory. Mirroring this division, we decompose our framework into long-term memory for instance association and short-term memory for instance update, realised by three lightweight yet synergistic modules: **1)** Long-term memory (LTM), detailed in Sec. 3.2, matches instance identities over extended periods, enabling recovery after prolonged occlusion. **2)** Short-term memory (STM), detailed in Sec. 3.3, recurrently updates each instance's representation with information from the immediately preceding frame. **3)** Spatial Consistency Learning (SCL) includes Learning-Based Mask Integration at inference and Instance-Consistency Mask Supervision during training, detailed in Sec. 3.4, respectively counteract VFM's intrinsic over-segmentation, thereby reducing query redundancy and furnishing STM and LTM with coherent, high-fidelity mask representations.

### 3.2 Long-Term Memory for Instance Association

Online 3D segmentation requires that all point-cloud observations of the same instance, collected across successive frames, be fused into a single temporally coherent instance. To improve the

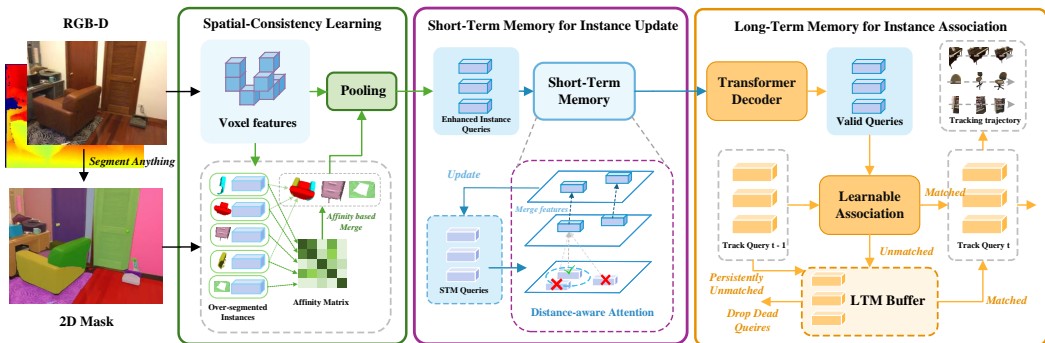

Figure 1: **This diagram delineates the operational mechanisms of our constituent modules.**
Spatial Consistency Learning (SCL) mitigates the over-segmentation tendencies of VFM by employ-
ing a one-to-many supervision strategy during the training phase and utilizing learning-based mask
integration at the inference stage. The Short-term Memory (STM) module enriches current instance
representations by integrating observational data from prior frames. Furthermore, the Long-term
Memory (LTM) module is engineered to associate instances, segmented by the Visual Front-end
Module (VFM), with established tracklets in memory, consequently enhancing temporal consistency.

temporal consistency, we recast instance aggregation as an explicit instance tracking problem with
supervised matching, confidence gating, and Hungarian assignment.

Concretely, at the first frame ($t = 1$), we obtain $N_1$ instance queries that derived from 3D mask
(Eq. 5) and their corresponding embeddings $\mathbf{Q}_1 \in \mathbb{R}^{N_1 \times d}$ that derived by applying a MLP to instance
queries, and predicted 3D bounding boxes $\mathbf{B}_1 \in \mathbb{R}^{N_1 \times 6}$. Each box is axis-aligned and specified
by its minimum and maximum coordinates ($x_{min}$, $y_{min}$, $z_{min}$, $x_{max}$, $y_{max}$, $z_{max}$). Similarly, for
every subsequent frame $t$ containing $N_t$ segments, we obtain the instance embeddings $\mathbf{Q}_t \in \mathbb{R}^{N_t \times d}$,
the corresponding boxes $\mathbf{B}_t \in \mathbb{R}^{N_t \times 6}$, and the instance embeddings from the tracklets in memory
$\mathbf{Q}^{Trk} \in \mathbb{R}^{N^{Trk} \times d}$. Here, $N^{Trk}$ represents the number of active tracklets up to now. Critically, the
embedding associated with each tracklet is not merely derived from the immediately preceding frame
$t - 1$, instead, it encapsulates richer temporal information accumulated across the sequence, thereby
reflecting a more comprehensive long-term history (see Sec. 3.3 for more details). Then we measure
the similarity between instances (segments) from current frames with tracklets by,

$$\mathbf{E}_{ij}^{app} = \mathbf{Q}_t[i] \odot \mathbf{Q}^{Trk}[j], \quad \mathbf{E}_{ij}^{geo} = \text{MLP}\big(\text{IoU}(\mathbf{B}_t[i], \mathbf{B}^{Trk}[j])\big), \quad \mathbf{E}_{ij} = \mathbf{E}_{ij}^{app} + \mathbf{E}_{ij}^{geo}, \quad (1)$$

where $\mathbf{E}^{app}, \mathbf{E}^{geo}, \mathbf{E} \in \mathbb{R}^{N_t \times N^{Trk} \times d}$ respectively correspond to the appearance, geometric, and
fused affinity features. We then project the fused affinity features using learned $\mathbf{w}, \mathbf{w}' \in \mathbb{R}^d$,

$$M_{ij} = \frac{\exp\big(\mathbf{w}^\top \mathbf{E}_{ij}\big)}{\sum_{j'} \exp\big(\mathbf{w}^\top \mathbf{E}_{ij'}\big)}, \quad C_{ij} = \sigma\big(\mathbf{w}'^\top \mathbf{E}_{ij}\big), \quad A_{ij} = M_{ij}\, C_{ij}. \quad (2)$$

where $M_{ij}$ is a row-normalised affinity obtained via softmax, which represents the relative similarity
that segmented instance $i$ corresponds to tracklet $j$. Since each row sums to 1, every segment allocates
its entire probability mass across the set of candidate tracklets. To modulate this raw affinity we
introduce a sigmoid-based confidence gate $C_{ij} = \sigma(\cdot)$, which down-weights uncertain matches and
suppresses spurious associations. To convert these probabilities into one-to-one correspondences, we
formulate a bipartite matching problem. Specifically, segments and tracklets form a bipartite graph
with edges weighted by gated affinities $A_{ij}$. Solving this assignment with the Hungarian algorithm
selects the set of pairs $(i, j)$ that maximises the summed weights while ensuring that every segment
and every track is used at most once. For each matched pair, the track state is updated by,

$$\mathbf{B}^{Trk}[j] \leftarrow \frac{\alpha_j \mathbf{B}^{Trk}[j] + \mathbf{B}_t[i]}{\alpha_j + 1}, \quad \mathbf{Q}^{Trk}[j] \leftarrow \frac{\alpha_j \mathbf{Q}^{Trk}[j] + \mathbf{Q}_t[i]}{\alpha_j + 1}, \quad \alpha_j \leftarrow \alpha_j + 1, \quad (3)$$

where $\alpha_j$ is the track age. Unmatched instances initialise new tracklets with $\alpha = 1$. Tracks that
remain unmatched for more than $T_{life}$ frames are marked stale, removed from the active set, and
pushed into a fixed-capacity queue, the oldest entry is evicted when the buffer overflows. At every
time step $t$, any segment that remains unassigned after the active-track matching stage is subsequently

matched against the LTM buffer with the identical Hungarian solver. A successful match reactivates the stored tracklet, restoring its state and resetting its age, thereby recovering instances that reappear after extended occlusion. By coupling confidence-gated Hungarian assignment with a bounded long-term memory, the proposed strategy suppresses spurious instances, maintains identities through prolonged occlusions, and guarantees constant computational overhead.

### 3.3 Short-Term Memory for Instance Update

In online 3D segmentation, the inherent scene continuity, where instances visible in frame $t-1$ often reappear in frame $t$, naturally motivates the use of cross-frame attention mechanisms to integrate historical context. For embodied agents, which frequently experience rapid and significant viewpoint variations, the effective fusion of object-level appearance information gathered from these diverse perspectives is particularly crucial. This capability to integrate multi-view object-centric features allows the agent to build a more robust and consistent understanding of instances over time, despite substantial changes in their observed appearance. Therefore, we design a instance update module that reuses and continually refines instance-centric embeddings.

We recognize that applying global cross-attention between all $N_t$ current queries and the instance embeddings from the previous frame can introduce substantial noise. This occurs because background queries often form irrelevant associations with prior instance features, thereby degrading the fusion process. To instill explicit instance awareness and mitigate this, we introduce the distance-aware Short-Term Memory (STM). Specifically, to filter out irrelevant interactions we adopt the distance-aware attention, which gates attention by Euclidean distance between instance centroids,

$$\text{Attn}(\mathbf{Q}'_t, \mathbf{K}_{t-1}, \mathbf{V}_{t-1}) = \text{Softmax}\Big(\frac{\mathbf{Q}'_t \mathbf{K}^{\top}_{t-1}}{\sqrt{d}} - \text{diag}(\boldsymbol{\tau}_t)\, \mathbf{D}^{(t-1,t)}\Big)\mathbf{V}_{t-1}, \tag{4}$$

where $\mathbf{Q}'_t \in \mathbb{R}^{N_t \times d}$ denotes current instance queries, the memory key $\mathbf{K}_{t-1} \in \mathbb{R}^{N_{t-1} \times d}$ and value $\mathbf{V}_{t-1} \in \mathbb{R}^{N_{t-1} \times d}$ are derived from $\mathbf{Q}_{t-1}$, $\mathbf{D}^{(t-1,t)} \in \mathbb{R}^{N_t \times N_{t-1}}$ stores pairwise centroid distances and $\boldsymbol{\tau}_t = [\tau_1, \ldots, \tau_{N_t}]^{\top}$ contains query-specific receptive-field scales. We predict these scales with a shared linear layer, $\boldsymbol{\tau}_t = \text{Linear}(\mathbf{Q}'_t)$, so each query adaptively narrows or widens its spatial scope. Large $\tau_i$ suppress attention to distant memory slots, encouraging local refinement, whereas small $\tau_i$ retain a global context when necessary. By suppressing attention to remote regions and modulating each query's receptive field, short-term memory yields temporally enhanced embeddings $\mathbf{Q}_t$.

### 3.4 Spatial Consistency Learning for Robust Association

As illustrated in Fig 2, VFMs like SAM [1] frequently fragments a single instance into several neighbouring masks. This fragmentation compromises effective cross-frame instance association. Previous methods [6] ignore this, resulting in degraded spatial coherence. To mitigate this gap, we introduce learning-based mask integration (LMI) at inference to merge high-affinity fragments and instance consistency mask supervision (ICMS) during training to apply one-to-many supervision.

**Learning-Based Mask Integration.** To recover coherent masks, we learn an affinity matrix that merges masks belonging to the same instance at every frame $t$. Given point cloud features $\mathbf{P}_t$ and corresponding 2D masks set $\mathcal{M}_t$, we can get query features $\mathbf{Q}_t$ and position $\mathbf{X}_t$ through

$$(\mathbf{Q}_t, \mathbf{X}_t) = \text{Pool}(\mathbf{P}_t, \mathcal{M}_t), \quad \mathcal{M}_t = \{\,\mathbf{m}_i\}_{i=1}^{N_t}, \tag{5}$$

where Pool aggregates point features within each mask. We then predicts axis-aligned bounding boxes $\mathbf{B}_t = \text{MLP}(\mathbf{Q}_t) \in \mathbb{R}^{N_t \times 6}$, since the boxes generated by corresponding 3D mask may not be a complete object [6]. For each pair $(i, j)$ we compute the affinity feature $\mathbf{E}_{ij}$ and $A_{ij}$ as in Sec. 3.2. We first perform hierarchical clustering to identify mask groups whose pairwise affinities $A_{ij}$ all exceed $\delta$. We then merge the masks within each such group to form $\tilde{\mathcal{M}}_t$, and finally re-pool features over these merged masks,

$$(\mathbf{Q}'_t, \mathbf{X}'_t) = \text{Pool}(\mathbf{P}_t, \tilde{\mathcal{M}}_t). \tag{6}$$

The mask-aggregation module is invoked only at inference. During training we intentionally refrain from merging masks to leverage fragment diversity as implicit data augmentation, detailed below.

**Instance Consistency Mask Supervision.** In addition to retaining fragmented masks as implicit data augmentation, each fragment can also provides a complementary view of the same object. Therefore,

we supervise corresponding fragments (many) with each ground-truth instance (one). This strategy improves robustness to low-quality masks, yields more consistent predictions for fragmented queries, and simplifies duplicate removal, which is vital for instance association in Sec. 3.2.

To formalise this supervision, consider a ground-truth instance $\mathbf{g}_k$ and its corresponding query set

$$\mathcal{Q}_k = \left\{ \mathbf{q}_i \mid \frac{|\mathcal{P}(\mathbf{m}_i) \cap \mathcal{P}(\mathbf{g}_k)|}{|\mathcal{P}(\mathbf{m}_i)|} > 0.5 \right\}, \tag{7}$$

where $\mathcal{P}(\cdot)$ denotes the pixel-set of a mask $\mathbf{m}_i$ and its corresponding query $\mathbf{q}_i$. We then enforce consistency across these fragments via

$$L_{1:N} = \sum_{k=1}^{N_{gt}} \sum_{\mathbf{q}_i \in \mathcal{Q}_k} \ell\big(f(\mathbf{q}_i), \mathbf{y}_k\big). \tag{8}$$

where $\ell$ is the loss and $\mathbf{y}_k$ denotes the ground-truth. However, as shown in Tab. 8, naively replacing the original one-to-one loss with $L_{1:N}$ leads to a consistent drop in segmentation accuracy. Our experiments demonstrate that this modification erodes the model's capacity to select the highest-quality fragment, which is supported by the self-attention mechanism. To satisfy both objectives, we configure the decoder in two distinct branches. In the first branch we enable self-attention and employ standard one-to-one supervision in order to preserve fragment selection capability. In the second branch we disable self-attention and apply one-to-many supervision in order to strengthen robustness across diverse fragments. Notably, this dual-branch mechanism is active only during training and incurs no additional computational cost at inference time.

### 3.5 Loss Functions

Our framework is trained end-to-end by minimising:

$$\mathcal{L} = \mathcal{L}_{seg} + \beta_{ltm}\, \mathcal{L}_{ltm} + \beta_{agg}\, \mathcal{L}_{agg}, \tag{9}$$

where the scalars $\beta_*$ weight the contribution of each term.

**Segmentation loss** $\mathcal{L}_{seg}$. The dual-decoder architecture described in Sec. 3.4 yields three sub-losses:

$$\mathcal{L}_{seg} = \mathcal{L}_{1:1} + \lambda\, \mathcal{L}_{1:N} + \gamma\, \mathcal{L}_{bg}, \tag{10}$$

where $\mathcal{L}_{1:1}$ enforces one-to-one assignment, $\mathcal{L}_{1:N}$ ensures consistency across masks through multi-target supervision , and $L_{bg}$ penalizes background masks.

**Long-term memory loss** $\mathcal{L}_{ltm}$. We introduce a matrix $y_{ij}$, where $y_{ij} = 1$ if query $q_i$ and track $t_j$ refer to the same ground-truth instance, and $y_{ij} = 0$ otherwise. We compute a one-to-one assignment $\pi^*$ by applying the Hungarian algorithm to the cost matrix $-\log \widehat{M}_{ij}$. The matching loss becomes

$$\mathcal{L}_{match} = -\frac{1}{N_t} \sum_{(i,j) \in \pi^*} \log \widehat{M}_{ij}. \tag{11}$$

To generate the sigmoid gate $C_{ij}$ for confidence we add

$$\mathcal{L}_{conf} = -\frac{1}{N_t N_{t-1}} \sum_{i,j} \big[ y_{ij} \log C_{ij} + (1 - y_{ij}) \log(1 - C_{ij}) \big]. \tag{12}$$

The full long-term memory loss is then

$$\mathcal{L}_{ltm} = \mathcal{L}_{match} + \beta_{conf}\, \mathcal{L}_{conf}, \tag{13}$$

**Mask-aggregation loss** $\mathcal{L}_{agg}$. To supervise the affinity predictor in LMI, we employ binary cross-entropy over positive pairs $\mathcal{P}$ and negative pairs $\mathcal{N}$:

$$\mathcal{L}_{agg} = -\frac{1}{|\mathcal{P}|} \sum_{(i,j) \in \mathcal{P}} \log A_{ij} - \frac{1}{|\mathcal{N}|} \sum_{(i,j) \in \mathcal{N}} \log\big(1 - A_{ij}\big). \tag{14}$$

Pairs whose masks overlap a ground-truth instance by over 50 % are positive, all others are negative.

Table 1: Class-agnostic 3D instance segmentation results of different methods on ScanNet200 dataset.

| Method | Present at | Type | VFM | AP | $AP_{50}$ | $AP_{25}$ | FPS |
|---|---|---|---|---|---|---|---|
| SAMPro3D [46] | 3DV'2025 | Offline | SAM | 18.0 | 32.8 | 56.1 | – |
| Open3DIS [47] | CVPR'2024 | Offline | GroundedSAM | 34.6 | 43.1 | 48.5 | – |
| SAI3D [22] | CVPR'2024 | Offline | SemanticSAM | 28.2 | 47.2 | 67.9 | – |
| SAM3D [19] | ICCVW'2023 | Online | SAM | 20.2 | 35.7 | 55.5 | 0.4 |
| ESAM [6] | ICLR'2025 | Online | SAM | 42.2 | 63.7 | 79.6 | 0.7 |
| **AutoSeg3D (Ours)** | - | Online | SAM | **45.5** | **66.7** | **81.0** | 0.7 |
| ESAM-E | ICLR'2025 | Online | FastSAM | 43.4 | 65.4 | 80.9 | 10.6 |
| **AutoSeg3D (Ours)** | - | Online | FastSAM | **46.2** | **67.9** | **81.7** | 10.1 |

Table 2: 3D instance segmentation results of different methods on ScanNet and SceneNN datasets. [*] denotes represent the results we reproduced following the official released config.

| Method | Present at | Type | ScanNet | | | SceneNN | | |
|---|---|---|---|---|---|---|---|---|
| | | | AP | $AP_{50}$ | $AP_{25}$ | AP | $AP_{50}$ | $AP_{25}$ |
| TD3D [48] | ICME'2024 | Offline | 46.2 | 71.1 | 81.3 | – | – | – |
| Oneformer3D [49] | CVPR'2024 | Offline | 59.3 | 78.8 | 86.7 | – | – | – |
| INS-Conv [38] | CVPR'2022 | Online | – | 57.4 | – | – | – | – |
| TD3D-MA [48] | ICME'2024 | Online | 39.0 | 60.5 | 71.3 | 26.0 | 42.8 | 59.2 |
| ESAM [6][*] | ICLR'2025 | Online | 41.6 | 59.6 | 75.2 | 30.3 | 47.6 | 63.4 |
| **AutoSeg3D (Ours)** | - | Online | **43.4** | **62.5** | **77.4** | **33.1** | **52.6** | **63.8** |

# 4 Experiments

## 4.1 Experiment Settings

Following our baseline ESAM [6], we begin by training a single-view perception model on ScanNet(200)-25k, a subset of ScanNet200 [7] with RGB-D frames. Then we fine-tune it on RGB-D sequences with full loss functions and randomly sample 8 RGB-D frames per scene at each training step. For the optimization settings, we use an AdamW optimizer with a learning rate of 0.0001 and a weight decay of 0.05 and the batch size is set to 4. All experiments are conducted using PyTorch on a single NVIDIA Tesla A100 GPU. Our experiments are conducted on ScanNet [8], ScanNet200 [7], SceneNN [9], and 3RScan [10] datasets.

## 4.2 Comparison with State-of-the-arts

**Results on ScanNet200 of Class-agnostic Setting.** Tab. 1 details the class-agnostic results on ScanNet200, demonstrating the superiority of our approach over existing state-of-the-art methods. Specifically, when SAM serves as the 2D segmentation model, our method achieves gains of 3.3 in AP, 3.0 in $AP_{50}$, and 1.4 in $AP_{25}$ compared to the recent ESAM [6]. The consistent performance improvements, even with a more lightweight 2D segmentation model such as FastSAM [50], underscore the effectiveness and generalizability of our method.

**Results on ScanNet and SceneNN.** Following the experimental setup of ESAM [6], Tab. 2 reports the results of our method, which is trained on ScanNet and subsequently evaluated on both ScanNet and SceneNN to assess its generalization performance. The notable improvements across multiple evaluation metrics and datasets strongly demonstrate the effectiveness and generalizability of our approach. Specifically, our method achieves significant gains of 1.8 in AP, 2.9 in $AP_{50}$, and 2.2 in $AP_{25}$ on ScanNet evaluation compared to ESAM.

**Results on SceneNN and 3RScan.** Tab. 3 reports the results of our method, trained on ScanNet200 and evaluated on SceneNN and 3RScan, which again demonstrate its strong generalization capabilities. Our approach surpasses previous methods, achieving significantly higher $AP_{50}$ and $AP_{25}$ scores. This underscores the effectiveness and adaptability of our method for robotic applications.

Table 3: Results of transferring different methods trained on ScanNet200 to SceneNN and 3RScan. "-E" indicates using FastSAM instead of SAM for 2D segmentation.

| Method | Present at | Type | ScanNet200→SceneNN | | | ScanNet200→3RScan | | |
|---|---|---|---|---|---|---|---|---|
| | | | AP | AP$_{50}$ | AP$_{25}$ | AP | AP$_{50}$ | AP$_{25}$ |
| SAMPro3D [46] | 3DV'2025 | Offline | 12.6 | 25.8 | 53.2 | 3.9 | 8.0 | 21.0 |
| Open3DIS [47] | CVPR'2024 | Offline | 18.2 | 32.2 | 48.9 | 9.5 | 21.8 | 47.0 |
| SAI3D [22] | CVPR'2024 | Offline | 18.6 | 34.7 | 65.7 | 8.1 | 16.9 | 37.0 |
| SAM3D [19] | ICCVW'2023 | Online | 15.1 | 30.0 | 51.8 | 6.2 | 13.0 | 33.9 |
| ESAM [6] | ICLR'2025 | Online | 28.8 | 52.2 | 69.3 | 14.1 | 31.2 | 59.6 |
| **AutoSeg3D (Ours)** | - | Online | **29.7** | **53.6** | **71.9** | **16.0** | **32.4** | **60.7** |
| ESAM-E [6] | ICLR'2025 | Online | 28.6 | 50.4 | 71.0 | 13.9 | 29.4 | 58.8 |
| **AutoSeg3D (Ours)** | - | Online | **30.2** | **54.1** | **72.8** | **16.8** | **34.3** | **61.0** |

## 4.3 Ablation Studies and Further Analysis

**Component-wise Ablation.** To further investigate the effects of our designs, we conduct an ablation studies on the ScanNet200. As depicted in Tab. 4, the introduction of long-term memory obtains gains of 2.5 and 2.9 in AP and AP$_{50}$ (② vs. ①), because of its effectiveness in instance association. The integrating of short-term memory enhances our model's ability to capture positional and content details from previous frames, resulting in performance improvements of 1.3 and 0.9 in AP and AP$_{50}$ respectively (③ vs. ①). ⑤ and ⑥ proves the effectiveness of components in the proved spatial consistency learning. The synergistic combination of all proposed elements constitutes an effective tracking-centric 3D segmentation framework, as in ⑦.

Table 4: Component-wise ablation.

| | LTM | STM | LMI | ICMS | AP | AP$_{50}$ | AP$_{25}$ |
|---|---|---|---|---|---|---|---|
| ① | – | – | – | – | 41.6 | 62.9 | 78.7 |
| ② | ✓ | – | – | – | 44.1 | 65.8 | 80.7 |
| ③ | – | ✓ | – | – | 42.9 | 63.8 | 80.0 |
| ④ | ✓ | ✓ | – | – | 44.8 | 66.7 | 81.0 |
| ⑤ | ✓ | ✓ | – | ✓ | 45.6 | 66.9 | 81.2 |
| ⑥ | ✓ | ✓ | ✓ | – | 45.5 | 67.0 | 81.3 |
| ⑦ | ✓ | ✓ | ✓ | ✓ | 46.2 | 67.9 | 81.7 |

**Long-Term Memory.** As shown in Tab. 5, compared to the baseline without LTM, adding geometric and appearance cues leads to steady gains (② vs. ①). Incorporating confidence estimation brings a notable boost, while further combining the recall mechanism achieves the highest scores, with AP increasing from 43.2 to 46.2 (+3.0), AP$_{50}$ from 64.8 to 67.9 (+3.1), and AP$_{25}$ from 80.4 to 81.7 (+1.3) (⑤ vs. ①). The recall and confidence strategies enable the model to effectively handle challenging cases such as long-term occlusions and ambiguous matches, resulting in more reliable temporal consistency throughout the sequence.

Table 5: Ablation for LTM.

| | Strategy | AP | AP$_{50}$ | AP$_{25}$ |
|---|---|---|---|---|
| ① | w/o LTM | 43.2 | 64.8 | 80.4 |
| ② | + Geometric | 43.7 | 65.4 | 80.5 |
| ③ | + Appearance | 45.0 | 66.5 | 81.3 |
| ④ | + Confidence | 45.5 | 67.2 | 81.5 |
| ⑤ | + Recall | 46.2 | 67.9 | 81.7 |

**Short-Term Memory.** As shown in Tab. 6, starting from the baseline without STM, solely introducing cross-frame attention brings limited improvement due to potential noise from irrelevant associations (② vs. ①). By further incorporating our distance-aware attention, which gates memory updates based on instance centroid distances, we observe a clear performance boost (③ vs. ①). Equipped with query-specific receptive-field scales, the final STM boosts AP from 44.7 to 46.2 (+1.5), AP$_{50}$ from 66.4 to 67.9 (+1.5), and AP$_{25}$ from 81.3 to 81.7 (+0.4) (④ vs. ①). These results demonstrate that explicitly modeling spatial proximity and adaptive receptive fields effectively suppresses noisy associations and enhances instance update accuracy in dynamic scenes.

Table 6: Ablation for STM.

| | Strategy | AP | AP$_{50}$ | AP$_{25}$ |
|---|---|---|---|---|
| ① | w/o STM | 44.7 | 66.4 | 81.3 |
| ② | + cross | 45.1 | 66.5 | 81.2 |
| ③ | + distance | 45.8 | 67.5 | 81.6 |
| ④ | + scale | 46.2 | 67.9 | 81.7 |

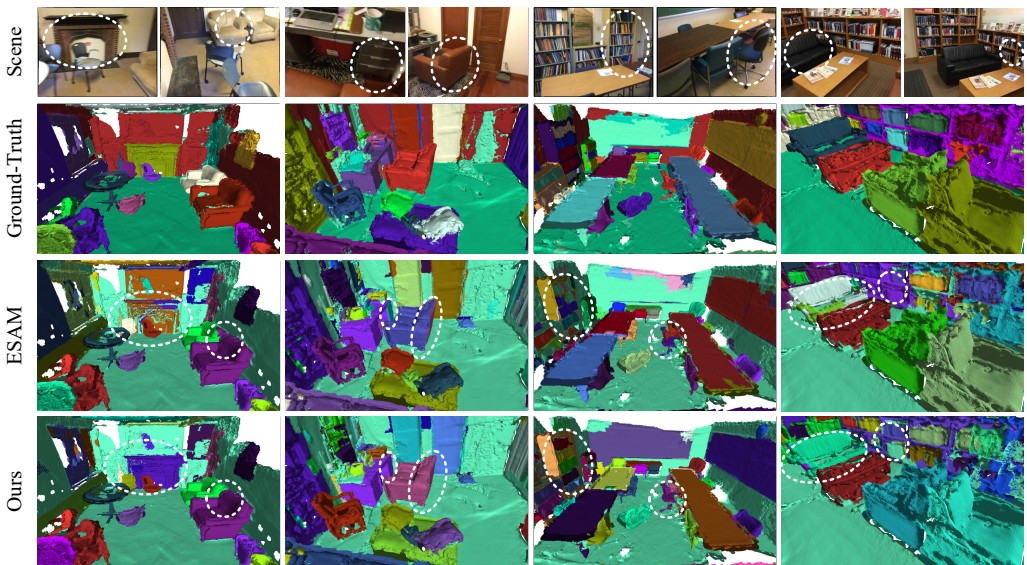

Figure 2: Visualization of segmentation results on ScanNet200 dataset.

Table 7: Ablation for LMI.

| | Strategy | AP | $AP_{50}$ | $AP_{25}$ |
|---|---|---|---|---|
| ① | w/o LMI | 45.6 | 66.9 | 81.0 |
| ② | train and infer. | 44.5 | 66.1 | 80.4 |
| ③ | infer. only | 46.2 | 67.9 | 81.7 |

**Learning-Based Mask Integration.** As shown in Tab. 7, applying Learning-Based Mask Integration (LMI) only during inference yields the best performance, improving AP from 45.6 to 46.2 (+0.6), $AP_{50}$ from 66.9 to 67.9 (+1.0) (③ $vs.$ ①). By contrast, incorporating LMI during training degrades performance, as early-stage inaccuracies introduce false mask fusions that hinder model convergence (② $vs.$ ①).

Table 8: Ablation of ICMS.

| | Strategy | TopK | AP | $AP_{50}$ | $AP_{25}$ |
|---|---|---|---|---|---|
| ① | w/o ICMS | – | 45.5 | 67.0 | 81.3 |
| ② | single-branch | 4 | 44.2 | 65.8 | 80.6 |
| ③ | dual-branch | 2 | 46.2 | 67.4 | 81.4 |
| ④ | dual-branch | 4 | 46.2 | 67.9 | 81.7 |
| ⑤ | dual-branch | 6 | 46.1 | 67.3 | 81.3 |
| ⑥ | dual-branch | 8 | 46.1 | 67.2 | 81.4 |

**Instance Consistency Mask Supervision.** As shown in Tab. 8, introducing Instance Consistency Mask Supervision (IMCS) with dual branches and TopK=4 achieves the best performance, improving AP from 45.5 to 46.2 (+0.7) (④$vs.$①). Here, setting K=4 indicating chosing the four masks exhibiting the highest similarity scores when compared to the ground truth of one object. By contrast, the single-branch configuration incurs a substantial drop in all metrics, underscoring the importance of combining one-to-one and one-to-many supervision signals.

**Qualitative Analysis.** We present a qualitative analysis conducted on the ScanNet validation set, with illustrative examples provided in Fig. 2. These results further substantiate the superior instance segmentation capabilities of our proposed model. The visualizations demonstrate that our model not only accurately segments target objects but also effectively rectifies over-segmented masks.

## 5 Conclusion and Limitation

**Conclusion.** In this paper, we present a novel, tracking-centric framework for online, real-time, and fine-grained 3D instance segmentation. By recasting the task as continuous instance tracking, our approach integrates Long-Term Memory for robust identity propagation, Short-Term Memory for immediate temporal context, and Spatial Consistency Learning to suppress over-segmentation. Extensive experiments on multiple benchmarks demonstrate that our lightweight system achieves state-of-the-art accuracy while maintaining real-time efficiency. **Limitation.** Both our and previous methods do not explicitly model relative motion of moving objects. We leave this for future research.

## Acknowledgements

This work was supported in part by the Natural Science Foundation of China (Grant No. 62036011, 62422317, U22B2056, 62192782, 62503323), the Beijing Natural Science Foundation (Grant No. JQ22014, L223003). The work of Weiming Hu was also supported in part by the Natural Science Foundation of China (Grant No. U2441241, U24A20331, 62202470, 62473363).

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
