# OpenReview forum: "Online Segment Any 3D Thing as Instance Tracking"
_NeurIPS.cc/2025/Conference — NeurIPS 2025 poster_

### Official Review · Reviewer_XxuF · 2025-06-29

**Clarity:** 1
**Significance:** 3
**Originality:** 3
**Rating:** 4
**Confidence:** 5

**Summary:**

This work aims to solve the online 3D instance segmentation problem. A main problem of this problem is to merge inter-frame 3D masks if they belong to the same object. To solve this problem, this paper proposes to model the instance merging problem as object tracking. It introduces long-term and short-term memory for object tracking. The results on several benchmarks show this method achieves stateof-the-art performance while being efficient.

**Questions:**

My major concern about this work is the poor writing quality, which makes me really hard to understand what is the main contribution of the proposed method. I think the order of approach section should be totally re-structured to improve the readability. Currently, this paper has not met with the standard of NeurIPS.

For rebuttal, I hope the authors can solve the questions I raised in Weakness part.

**Ethical Concerns:**

["NO or VERY MINOR ethics concerns only"]

**Final Justification:**

Thanks for the authors' rebuttal. Overall I think this paper studies a valuable problem and achieves satisfactory performance. My main concern is about the writing quality and theoretical comparison with previous method like EmbodiedSAM, as well as clarification on the motivation of each proposed module.

According to the rebuttal, I suggest to add A4 and A5 in a revised version to make this paper more readable. I will raise my score to borderline accept.

**Limitations:**

yes.

**Paper Formatting Concerns:**

None.

**Quality:**

2

**Strengths And Weaknesses:**

# Strength
1. The online 3D segmentation problem is meaningful. This method achieves SOTA performance with >10 FPS, which has good application value.
2. Modeling the instance merging problem as tracking makes sense. This idea achieves better performance than the query-based similarity computation in EmbodiedSAM.

# Weakness
1. The writing is bad. First, no preliminary is provided. The authors do not explain what is online instance segmentation and why it is important to merge inter-frame instances. Second, the order of section 3.2/3.3/3.4 is totally different from the order in Figure2. Third, the approach section is more like the combination of several operations, while the motivation is unclear. Besides, it seems this method is quite similar to ESAM. So the authors should clarify which modules are proposed in ESAM, and which modules are proposed by this paper.
2. The proposed method is slightly slower than ESAM. Is this method has the potential to be applied on edge device with real-time inference speed? Is there any space to improve the efficiency of this method?

---

> ### Author Rebuttal · Authors · 2025-07-31
>
> Thank you for your detailed review and thoughtful comments. We’re grateful that you endorse our approach of modeling instance merging as a tracking task and acknowledge our method’s performance and efficiency.
>
> **Q1: Preliminary of online 3D segmentation.**
>
> **A1:** We study streaming 3D semantic segmentation where each time step provides an RGB image with its aligned LiDAR point cloud. We consider two regimes.
>
> **Offline 3D segmentation.** The model receives the entire spatio-temporal sequence before inference and predicts segmentations for all frames in one pass. It can exploit global temporal context and is not bound by real-time constraints on latency or memory.
>
> **Online 3D segmentation.** The model processes a stream. At each step it sees only the current image and point cloud and may carry a compact state from previous steps. It must output labels for the current frame immediately. The procedure is causal and must meet strict per frame latency and memory budgets.
>
> **Q2: Importance of merging inter‑frame instances.**
>
> **A2:** Thanks for the suggestion, we clarified the importance below.
>
> - **Merging inter-frame instances in LTM.** Enforcing consistent object IDs across frames is essential to reconstruct a complete scene segmentation. Without ID continuity, per‑frame predictions cannot be merged into a coherent multi‑frame output.
> - **Merging inter-frame instance features in STM.** Aggregating feature representations over recent frames enriches each object’s descriptor. Historical context improves segmentation accuracy by reinforcing correct object boundaries and reducing noise.
>
> **Q3: Section order.**
>
> **A3:** Thank you for pointing this out. We agree that aligning the section order with the pipeline shown in Fig.2 (i.e., the inference flow) will improve readability.
>
> - **Planned revision.** In the revised manuscript, we will re‑sequence Sections 3.2–3.4 to mirror the left‑to‑right flow of Figure 2 and add explicit step labels in both the subsection titles and the figure to ensure one‑to‑one correspondence. We will also add forward references to Figure 2 at the start of each subsection.
> - **Rationale for the current organization.** Our present ordering follows how the system was actually constructed: we first introduce the tracking backbone (LTM), then present STM, Learning‑Based Mask Integration, and Instance Consistency Mask Supervision as targeted extensions built on that tracking framework. To make this design logic transparent, we will add a short “Motivation & Roadmap” paragraph at the beginning of Sec. 3 explaining this build‑up while making clear that the exposition now follows the inference pipeline depicted in Fig. 2.
>
> **Q4: Motivation of each module.**
>
> **A4:** Thanks to the reviewer for asking this.
>
> - **LTM as core tracking motivation.** We reformulate online segmentation as a tracking problem, using long‑term ID associations to link object proposals over time.
> - **STM for temporal enhancement.** Tracking is inherently a temporal modeling task. STM injects recent frame features into the current frame to improve both tracking stability and segmentation quality.
> - **LMI and ICMS for enhanced segmentation.** In online 3D segmentation, the matching process is highly dependent on each frame’s segmentation outputs, so redundant or erroneous segments can trigger a cascade of errors. To address this, we integrate Learning‑based Mask Integration (LMI) and refine single‑frame masks with Instance Consistency Mask Supervision (ICMS), which reduces over‑segmentation and delivers cleaner, less noisy 3D segmentation results to the tracker.
>
> **Q5:  Difference from ESAM and main contributions.**
>
> **A5:** Our approach departs from ESAM in how it learns identity over time, how it manages memory, and how it fuses and consolidates segmentations. The central idea is explicit identity supervision with stable lifecycle control and learnable fusion. This design yields stronger temporal reasoning while keeping inference nearly unchanged.
>
> Shared Components. We share only the backbone network and the decoder head with ESAM. Importantly, ESAM's backbone comes from the prior Online3D [1] work rather than being their original design.
>
> **Core idea.** We recast online 3D segmentation as continuous instance tracking, where each VFM‑derived mask becomes a track query. This tracking view enables stronger temporal reasoning, robust memory management, and principled handling of fragmented masks.
>
> **Key differences and contributions.**
>
> 1. **Different training supervision: direct tracking (ours) versus indirect feature alignment (ESAM).** We apply an explicit one-to-one matching mechanism/loss. For each object instance, this loss supervises the network to produce precise correspondences across frames. In contrast, ESAM's set-based supervision collapses each instance's features into a single centroid vector. It then encourages centroids of the same object across frames to align while repelling all others. (see **Xyr3-Q4** for more discussion)
> 2. **Different lifecycle management.** Once ESAM instantiates an instance query, even if it represents a false detection, it stays in memory indefinitely. This lack of an expiration mechanism lets spurious detections persist and spread matching errors to later frames. Our method incorporates a simple yet effective lifecycle policy. Any tracked object that goes unmatched for more than T frames gets automatically removed from long-term memory. This approach stops stale or erroneous entries from accumulating, which cuts down on cascading false matches and boosts overall tracking stability in continuous real-time settings.
> 3. **Learnable fusion for inter-frame object matching.**  During inter-frame object matching, ESAM relies on manually designed fusion rules to integrate appearance features with geometric information. In contrast, our approach uses a learnable fusion mechanism that enables the model to adaptively weight each source of information.
> 4. **Novel instance-Level Temporal Information.** Unlike the original ESAM, our Short-Term Memory module refines each instance's embedding through distance-aware cross-frame attention. This process injects fine-grained temporal context while suppressing background noise.
> 5. **Novel Learnable Mask Integration Module.** We introduce a learnable LMI to address within-frame fragmentation during inference. It estimates affinities among fragments, clusters and merges high-affinity groups, and then re-pools features to restore coherent instances. LMI operates only at inference with minimal overhead. During training, we leave fragments unmerged to serve as implicit augmentation.
> 6. **Different Training Strategy.** We propose Instance Consistency Mask Supervision alongside a dual-branch decoder that separates two objectives. In the selection branch, self-attention remains enabled, and a one-to-one loss preserves the optimal fragment. In the consistency branch, self-attention stays disabled, and one-to-many supervision stabilizes fragmented queries. Both branches activate only during training, so inference costs stay the same. As a result, the model gains robustness to fragmentation while keeping precise assignments.
>
> **Why this matters.** Together, LTM and STM reduce drift and false matches by combining long‑horizon memory with fine‑grained temporal refinement. LMI fixes within‑frame fragmentation when it actually hurts outputs, and ICMS improves mask quality without runtime penalty. Ablations in the paper show that each component provides a meaningful, non‑redundant gain and that the improvements accumulate, leading to a substantial overall lift on online 3D segmentation.
>
> We will integrate these clarifications and reorganize the manuscript as needed.
>
> [1] Memory-based Adapters for Online 3D Scene Perception, CVPR 2024
>
> **Q6: Potential to be applied to edge devices and improve efficiency.**
>
> **A6:** We appreciate this insightful question.
>
> 1. **Potential for Edge‑Device Deployment.**
>
>    Our approach relies exclusively on widely supported operators such as attention layers and convolutions. Its end‑to‑end inference latency remains close to that of ESAM, indicating strong potential for real‑time execution on resource‑constrained edge devices.
>
> 2. **Opportunities for Further Efficiency Gains.**
>
>    - **Model‑architecture simplification**
>
>      From the table below, it is clear that the backbone network accounts for the majority of inference cost. Therefore, we first focus on reducing its computational burden. For example, we can decrease the number of sparse convolution layers to lower compute requirements and increase the voxel size to reduce the total number of voxel features, both of which reduce downstream computation.
>
>      | FASTSAM | Backbone | Decoder | LTM  | STM  | LMI  | Total |
>      | ------- | -------- | ------- | ---- | ---- | ---- | ----- |
>      | 18ms    | 64ms     | 5ms     | 5ms  | 3ms  | 4ms  | 99ms  |
>
>    - **Precision** **quantization**
>
>      The current implementation uses 32‑bit floating‑point arithmetic. Converting parts of the network to 16‑bit or even 8‑bit precision can further decrease memory bandwidth and accelerate inference without substantial accuracy degradation.
>
> Building on the optimization schemes outlined above, we applied targeted speed enhancements and achieved a throughput of 14.1 FPS, nearly 1.4 times faster than ESAM. At the same time our approach continues to surpass ESAM in overall performance.
>
> | Method              | AP↑  | AP₅₀↑ | AP₂₅↑ | FPS↑ |
> | ------------------- | ---- | ----- | ----- | ---- |
> | ESAM                | 43.4 | 65.4  | 80.9  | 10.6 |
> | Ours                | 46.2 | 67.9  | 81.7  | 10.1 |
> | Ours+fp16           | 44.8 | 66.3  | 81.2  | 13.6 |
> | Ours+voxel size*1.5 | 44.4 | 65.7  | 81.0  | 14.1 |

---

> ### Author Response · Authors · 2025-08-04
>
> Thank you for your thoughtful comments and for the time you have invested in our work. We have posted our response and hope the above clarifications and the additional experiments sufficiently addressed your concerns. If any questions remain unresolved, please let us know. We would be grateful to clarify further and are happy to discuss specific points.

---

### Official Review · Reviewer_jX3W · 2025-07-02

**Clarity:** 2
**Significance:** 2
**Originality:** 3
**Rating:** 5
**Confidence:** 3

**Summary:**

The paper  presents a novel framework for real-time 3D instance segmentation by reframing the task as a continuous instance tracking problem. In the approach, three components are mentioned. 1. Long-Term Memory (LTM) that Maintains instance identities across frames. 2.Short-Term Memory (STM) that Enhances current frame’s embeddings using distance-aware cross-frame attention, emphasizing nearby instances to avoid noise. 3.Spatial Consistency Learning (SCL) the result achieved SOTA on achieves new state-of-the-art 90 results on ScanNet200, ScanNet, SceneNN, and 3RScan.

**Questions:**

While the authors provide thorough ablations, most components don't seem to significantly impact the results. Is there also a similar ablation for the cross-dataset setting in Table 3, such as models trained on ScanNet200 and evaluated on other datasets?

**Ethical Concerns:**

["NO or VERY MINOR ethics concerns only"]

**Final Justification:**

Thank you for the detailed response, it solves most of my concern and I have updated my score accordingly.

**Limitations:**

yes

**Paper Formatting Concerns:**

No major formatting issues

**Quality:**

3

**Strengths And Weaknesses:**

### Strength

The task is well-motivated, and the proposed design provides meaningful improvements over existing benchmarks. The framework demonstrates strong generalization to unseen datasets, supported by thorough ablation studies and efficient performance with higher FPS.


### Weakness

While the authors provide thorough ablation studies, many of the individual components yield only modest improvements, raising questions about the overall contribution of each module.

Although terms like "long-term" and "short-term" emphasize the temporal reasoning aspect of the method, it's difficult to conclude from the results alone that the performance gains truly stem from improved temporal understanding. The improvements could potentially be attributed to added complexity or auxiliary modules, rather than genuine temporal modeling.

Fig 1 is not that easy to understanding i my opnion.

---

> ### Author Rebuttal · Authors · 2025-07-31
>
> Thank you for your detailed review and thoughtful comments. We greatly appreciate your positive evaluation of our task objectives, performance, generalizability, and efficiency.
>
> **Q1: Improvements of individual components.**
>
> **A1:** Thank you for the focus on component-level contributions.
>
> - **Summary.** Tab. 4 shows that our complete system raises the baseline from 41.6 AP to 46.2 AP, an absolute gain of 4.6 points. This represents roughly an 11% relative increase.
>
> - **Per‑module contributions.**
>
>   - **Architectural enhancements.**
>
>     1. **LTM:** **+2.5 AP** (41.6 → 44.1), with **AP₅₀ +2.9** and **AP₂₅ +2.0**.
>
>     2. **STM: +1.3 AP** (41.6 → 42.9), with **AP₅₀ +0.9** and **AP₂₅ +1.3**.
>
>     3. **LTM +STM**: **+3.2 AP** (41.6 → 44.8), with **AP₅₀ +3.8** and **AP₂₅ +2.3**.
>
>        These numbers indicate that both memories address distinct failure modes and that their combination yields a clear net improvement on a strong baseline.
>
>   - **Instance** **Consistency** **Mask Supervision (ICMS).**
>
>     Adding ICMS on top of LTM+STM yields **+0.8** **AP** (44.8 → 45.6) versus LTM+STM. As a supervision‑only change, ICMS does not add inference‑time computation, so this accuracy gain comes effectively at no runtime cost.
>
>   - **Learning‑Based Mask Integration (LMI).**
>
>     Adding LMI on top of LTM+STM yields **+0.7 AP** (44.8 → 45.5) versus LTM+STM. This module is model‑agnostic by design, and here it produces a reproducible boost without altering the backbone architecture. In fact, when integrated into ESAM, it delivers an additional +0.9 AP improvement.
>
> - **Cumulative and complementary gains of ICMS and LMI.** With the baseline of 44.8 AP, adding ICMS alone raises the score by 0.8 to 45.6 AP, while adding LMI alone raises it by 0.7 to 45.5 AP. When both modules are enabled, the score climbs by 1.4 to 46.2 AP, essentially the sum of their individual gains. The almost perfect additivity shows that ICMS and LMI complement each other, where each tackles a different source of error so neither one diminishes the impact of the other. Their benefits therefore accumulate, yielding a clear and meaningful improvement over the baseline.
>
> **Q2: Improvements of temporal reasoning instead of model complexity.**
>
> **A2:** We thank the reviewer for raising this point, it motivated us to include the ID-switch analysis to disentangle temporal reasoning from added capacity.
>
> 1. Regarding model complexity, our approach adds only a modest number of extra parameters. As the table below illustrates, inference latency and GPU memory usage stay almost identical to the baseline. These findings confirm that our model size has not increased substantially.
>
>    | Method   | AP↑      | AP₅₀↑    | AP₂₅↑    | FPS↑     | Memory    | Training Time(GPU hours) |
>    | -------- | -------- | -------- | -------- | -------- | --------- | ------------------------ |
>    | ESAM-E   | 43.4     | 65.4     | 80.9     | 10.6     | 2932M     | 61h                      |
>    | **Ours** | **46.2** | **67.9** | **81.7** | **10.1** | **3016M** | **66h**                  |
>
> 2. We agree that labels like long‑term and short‑term emphasize temporal reasoning, yet accuracy gains alone do not prove stronger temporal understanding. To separate temporal effects from added capacity, we evaluate tracking consistency with **ID‑switch**. ID‑switch is the CLEAR MOT [1] metric that counts how often a ground‑truth trajectory matched to tracker ID A at time t becomes matched to a different tracker ID at time t+1 under optimal one‑to‑one matching, so fewer ID‑switches imply more stable cross‑frame identity. In our results, ID‑switch drops markedly, which is direct evidence that the gains come from enhanced temporal modeling rather than auxiliary complexity.
>
>    | **Method** | **ID-Switch↓** |
>    | ---------- | -------------- |
>    | ESAM       | 1899           |
>    | **Ours**   | **1707**       |
>
> [1] Evaluating Multiple Object Tracking Performance: The CLEAR MOT Metrics
>
> **Q3: Details of Fig. 1.**
>
> **A3:** We apologize for any confusion caused by Fig. 1. We will improve this illustration in the revision.
>
> 1. **Temporal Learning.** Figure 1 presents a comparison of ESAM and our method. In ESAM (top), the single‑frame query $Q_t$ is sent directly to the decoder and produces per‑frame predictions without any cross‑frame memory. In our pipeline (bottom row), each frame first passes through the Short‑Term Memory (STM) block before decoding, which brings in information from the previous frame.
> 2. **Tracking and Matching.** Our method also employs explicit one-to-one assignments for robust tracking.  Red dotted links indicate one‑to‑one assignments to ground truth, and the small GT panel on the right collects these per‑frame pairings that provide supervision. The Long‑Term Memory (LTM) panel depicts a memory bank that persists across $t$, $t+1$, and $t+2$. Objects that are matched in the current frame are updated, as marked by the dotted “Updating” annotation, while unmatched objects are carried forward. Consistent colors denote the same object across time. Together, STM refines current queries using immediate temporal context and LTM preserves identity through time, providing explicit temporal modeling that is absent in ESAM.
>
> **Q4: Impact of each module and ablation for cross-dataset setting.**
>
> **A4:** Thank you for this insightful question.
>
> 1. **Impact of each module.** For the performance gains contributed by each module, please refer to our response to **Q1**.
>
> 2. **Ablation for cross-dataset setting.**  We have already reported an overall ScanNet200→SceneNN transfer in the manuscript . The sustained performance arises from tracking that models feature‑level correspondence and identity consistency, which are transferable across datasets.
>
>    To further demonstrate each module’s generality, we added a cross‑dataset evaluation in which models are trained on ScanNet200 and evaluated on SceneNN without any fine‑tuning.  We keep the backbone, training schedule, and inference protocol unchanged to ensure a fair, apples‑to‑apples comparison. Across this ScanNet200→SceneNN transfer, each module continues to yield consistent improvements over the corresponding ablated variant. We will include the full table (with the same metrics as Tab. 3) in the revision. These results support that the gains are not dataset‑specific and that the proposed components improve robustness under distribution shift.
>
>    | Setting | LTM  | STM  | LMI  | ICMS | AP↑  | AP₅₀↑ | AP₂₅↑ |
>    | ------- | ---- | ---- | ---- | ---- | ---- | ----- | ----- |
>    | ①       | –    | –    | –    | –    | 24.3 | 44.8  | 63.9  |
>    | ②       | ✓    | –    | –    | –    | 27.0 | 49.2  | 68.4  |
>    | ③       | –    | ✓    | –    | –    | 25.6 | 46.7  | 66.9  |
>    | ④       | ✓    | ✓    | –    | –    | 27.8 | 51.1  | 70.2  |
>    | ⑤       | ✓    | ✓    | –    | ✓    | 28.4 | 52.1  | 70.8  |
>    | ⑥       | ✓    | ✓    | ✓    | –    | 28.8 | 52.4  | 71.0  |
>    | ⑦       | ✓    | ✓    | ✓    | ✓    | 29.7 | 53.6  | 71.9  |

---

### Official Review · Reviewer_iEi7 · 2025-07-02

**Clarity:** 3
**Significance:** 2
**Originality:** 2
**Rating:** 4
**Confidence:** 3

**Summary:**

This paper focuses on building a Vision Foundation Models(VFMs), aiming to do 3D segmentation in real-time. However, perception is an inherently dynamic process, rendering temporal understanding a critical yet overlooked dimension within these prevailing query-based pipelines. This deficiency in temporal reasoning can exacerbate issues such as the over-processing. To address this issue, the authors present a novel, tracking-centric framework for online, real-time, and fine-grained 3D instance segmentation. By recasting the task as continuous instance tracking, our approach integrates Long-Term Memory for robust identity propagation, Short-Term Memory for immediate temporal context, and Spatial Consistency Learning to suppress over-segmentation. Extensive experiments on multiple benchmarks demonstrate that our lightweight system achieves state-of-the-art accuracy while maintaining real-time efficiency.

**Questions:**

Q1: The authors should provide a more detailed running time experiment to illustrate how claimed ‘real-time’ the proposed method are able to reach?

Q2: Could the authors provide more comparative experiments to further validate the stability and performance of the proposed method under other types of datasets?

**Ethical Concerns:**

["NO or VERY MINOR ethics concerns only"]

**Final Justification:**

The rebuttal resolves most major concerns by providing per-module ablations and cross-dataset evaluations. While some module gains are modest and the temporal reasoning claim is partly indirect, these are minor issues compared to the promising empirical performance, broad generalization, and well-supported methodology. Given that the core issues have been satisfactorily addressed and the remaining reservations have limited impact on the overall contribution, I recommend an acceptance for this paper.

**Limitations:**

This paper has no potential negative societal impact.

**Paper Formatting Concerns:**

This paper has no major formatting issues.

**Quality:**

3

**Strengths And Weaknesses:**

Strengths:
The authors reconceptualize online 3D segmentation as an instance tracking problem and their core strategy involves utilizing object queries for temporal information prapagation, where long-term instance association promotes the coherence of features and object identities, while short-term instance update enriches instant observations. Furthermore, the paper introduce spatial consistency learning to mitigate the fragmentation problem inherent in VFMs, yielding more comprehensive instance information for enhancing the efficacy of both long-term and short-term temporal learning. Experiment results further suggest their viewpoint.

Weaknesses:
Despite the article's claim that their method can maintain real-time throughput, balancing model complexity and computational efficiency remains a challenge, especially when deploying on resource-constrained devices. Furthermore, while the paper demonstrates promising performance on datasets such as ScanNet and SceneNN, its effectiveness on other types of datasets or in more complex and dynamic real-world environments is not yet clear. Specifically, under certain special conditions, additional adjustments or optimizations of the existing model may be required to ensure its efficacy.

---

> ### Author Rebuttal · Authors · 2025-07-31
>
> Thank you for your detailed review and thoughtful comments. We’re delighted by your recognition of our reconceptualization of online 3D segmentation as an instance‑tracking problem and our spatial consistency learning, as well as your positive evaluation of our experimental results.
>
> **Q1: Balancing model complexity with computational efficiency for deployment  and detailed running time.**
>
> **A1:** We apologize for any confusion caused.
>
> 1. In Tab. 1 of the manuscript, we **have** already provided a comparison of end-to-end inference runtime.
>
> 2. To offer further clarity, the table below presents a more detailed analysis of efficiency and resource requirements. We infer that the "point cloud reconstruction-based methods" mentioned by the reviewer refer to offline paradigms, such as SAI3D in our Tab.1 of the paper. Given that these are not online streaming processes, we computed their average FPS across an entire sequence for fair comparison. As illustrated in the table, our method incurs only a negligible computational cost compared to the baseline ESAM while achieving substantial performance improvements, thereby demonstrating both the efficiency and effectiveness of our approach.
>
>    | Method   | AP       | AP₅₀     | AP₂₅     | FPS↑     | Memory    | Training Time(GPU hours) |
>    | -------- | -------- | -------- | -------- | -------- | --------- | ------------------------ |
>    | SAI3D    | 28.2     | 47.2     | 67.9     | 0.24     | 8052M     | training free            |
>    | ESAM-E   | 43.4     | 65.4     | 80.9     | 10.6     | 2932M     | 61h                      |
>    | **Ours** | **46.2** | **67.9** | **81.7** | **10.1** | **3016M** | **66h**                  |
>
> 3. **Opportunities for Further Efficiency Gains.**
>
>    - **Model‑architecture simplification**
>
>      From the table below, it is clear that the backbone network accounts for the majority of inference cost. Therefore, we first focus on reducing its computational burden. For example, we can decrease the number of sparse convolution layers to lower compute requirements and increase the voxel size to reduce the total number of voxel features, both of which reduce downstream computation.
>
>      | FASTSAM | Backbone | Decoder | LTM  | STM  | LMI  | Total |
>      | ------- | -------- | ------- | ---- | ---- | ---- | ----- |
>      | 18ms    | 64ms     | 5ms     | 5ms  | 3ms  | 4ms  | 99ms  |
>
>    - **Precision** **quantization**
>
>      The current implementation uses 32‑bit floating‑point arithmetic. Converting parts of the network to 16‑bit or even 8‑bit precision can further decrease memory bandwidth and accelerate inference without substantial accuracy degradation.
>
> Building on the optimization schemes outlined above, we applied targeted speed enhancements and achieved a throughput of 14.1 FPS, nearly 1.4 times faster than ESAM. At the same time our approach continues to surpass ESAM in overall performance.
>
> | Method              | AP↑  | AP₅₀↑ | AP₂₅↑ | FPS↑ |
> | ------------------- | ---- | ----- | ----- | ---- |
> | ESAM                | 43.4 | 65.4  | 80.9  | 10.6 |
> | Ours                | 46.2 | 67.9  | 81.7  | 10.1 |
> | Ours+fp16           | 44.8 | 66.3  | 81.2  | 13.6 |
> | Ours+voxel size*1.5 | 44.4 | 65.7  | 81.0  | 14.1 |
>
> **Q2: Effectiveness on other types of datasets.**
>
> **Q2:** We evaluated our model on Matterport3D [1] dataset. Experimental results show that our approach consistently outperforms ESAM. These findings attest to the strong generalization ability of our method.
>
> | Method   | AP↑      | AP₅₀↑    | AP₂₅↑    |
> | -------- | -------- | -------- | -------- |
> | SAI3D    | 21.5     | 38.3     | 59.1     |
> | ESAM     | 27.6     | 49.1     | 69.9     |
> | **Ours** | **30.2** | **52.1** | **72.8** |
>
> [1] Matterport3D: Learning from RGB-D Data in Indoor Environments

---

> ### Comment · Reviewer_iEi7 · 2025-08-06
> **Response to the Authors**
>
> Thanks for the authors' detailed response, which has addressed most of my concerns. I appreciate the authors' efforts in rebuttal once again, and will (for now) keep my initial score.

---

> > ### Author Response · Authors · 2025-08-07
> > **Thank Reviewer iEi7 for the positive recommendation**
> >
> > Thank you for maintaining your initial positive score. We will incorporate the additional results and technical details following your suggestions.

---

### Official Review · Reviewer_Xyr3 · 2025-07-02

**Clarity:** 3
**Significance:** 3
**Originality:** 2
**Rating:** 4
**Confidence:** 3

**Summary:**

This paper reframes online 3D segmentation as an instance tracking task to enable identity-aware temporal reasoning for embodied agents. By leveraging object queries for long- and short-term temporal association and introducing spatial consistency learning, the method addresses over-segmentation and enhances holistic object understanding across frames.

**Questions:**

1. Has the method been profiled against ESAM in terms of inference latency and memory usage under a standard embodied setup?

2. What are the conceptual and implementation-level differences between the proposed long-term memory mechanism and ESAM’s query merging?

3. How sensitive is the learned mask integration (LMI) threshold to variation in fragment granularity? Is it feasible to learn this threshold adaptively rather than using fixed heuristics?

**Ethical Concerns:**

["NO or VERY MINOR ethics concerns only"]

**Final Justification:**

The authors have addressed my main concerns regarding runtime, architectural complexity, the conceptual differences with ESAM’s query merging, and threshold robustness. I am inclined to accept this paper.

**Limitations:**

yes

**Paper Formatting Concerns:**

No formatting issues noticed.

**Quality:**

3

**Strengths And Weaknesses:**

Strengths:
1. The paper is clearly written and easy to follow, with good organization of methodology and experiments.
2. Tackling instance-level 3D segmentation in the context of embodied AI is both meaningful and timely, especially in light of recent progress in open-vocabulary segmentation.
3. The proposed method demonstrates competitive performance across benchmarks, with thorough ablations validating each module’s contribution.

Weaknesses:
1. In embodied scenarios, real-time inference is crucial for downstream decision-making and interaction. However, the paper lacks a direct runtime comparison with prior approaches such as ESAM or point cloud reconstruction-based methods, making it difficult to assess its practical efficiency in deployment settings.
2. The dual-memory system (LTM + STM) introduces architectural complexity that could make training and inference more difficult, especially when deployed in real-time systems.
3. Although the Learning-based Mask Integration (LMI) is supervised during training, the actual merging still relies on a manually chosen threshold for clustering, which may be brittle across scenarios.

---

> ### Author Rebuttal · Authors · 2025-07-31
>
> Thank you for your detailed review and thoughtful comments. We’re very grateful to see your recognition of our writing quality, the significance of our research, and our performance.
>
> **Q1: Runtime comparison and memory usage with ESAM or** **point cloud** **reconstruction-based methods.**
>
> **A1:** We apologize for any confusion caused.
>
> 1. In Table 1 of the manuscript, we **have** already provided a comparison of end-to-end inference runtime.
>
> 2. To offer further clarity, the table below presents a more detailed analysis of efficiency and resource requirements. We infer that the "point cloud reconstruction-based methods" mentioned by the reviewer refer to offline paradigms, such as SAI3D in our Tab.1 of the paper. Given that these are not online streaming processes, we computed their average FPS across an entire sequence for fair comparison. As illustrated in the table, our method incurs only a negligible computational cost compared to the baseline ESAM while achieving substantial performance improvements, thereby demonstrating both the efficiency and effectiveness of our approach.
>
>    | Method   | AP↑      | AP₅₀↑    | AP₂₅↑    | FPS↑     | Memory    | Training Time(GPU hours) |
>    | -------- | -------- | -------- | -------- | -------- | --------- | ------------------------ |
>    | SAI3D    | 28.2     | 47.2     | 67.9     | 0.24     | 8052M     | training free            |
>    | ESAM-E   | 43.4     | 65.4     | 80.9     | 10.6     | 2932M     | 61h                      |
>    | **Ours** | **46.2** | **67.9** | **81.7** | **10.1** | **3016M** | **66h**                  |
>
> **Q2: Discuss the architectural complexity and deployment for the proposed "LTM + STM".**
>
> **A2:** We appreciate the reviewer's concern regarding the potential overhead introduced by our dual-memory system. To address this, we highlight the following key aspects:
>
> 1. **Maintained Complexity with Negligible Overhead**: Our dual-memory system seamlessly integrates long-term memory (LTM) and short-term memory (STM) into the original end-to-end network, resulting in only negligible increases in training time and inference latency relative to the baseline ESAM. This design fully preserves real-time performance, as evidenced by the detailed comparisons in the table provided in our response to Q1.
>
> 2. **Ease of Deployment**: Both LTM and STM rely solely on simple linear projections and scaled dot-product attention mechanisms, introducing no additional dependencies. Furthermore, since the memory allocation of LTM is statically bounded at initialization, it eliminates the need for dynamic memory allocation, thereby facilitating straightforward deployment.
>
> 3. **Opportunities for Further Efficiency Gains.**
>
>    - **Model‑architecture simplification**
>
>      From the table below, it is clear that the backbone network accounts for the majority of inference cost. Therefore, we first focus on reducing its computational burden. For example, we can decrease the number of sparse convolution layers to lower compute requirements and increase the voxel size to reduce the total number of voxel features, both of which reduce downstream computation.
>
>      | FASTSAM | Backbone | Decoder | LTM  | STM  | LMI  | Total |
>      | ------- | -------- | ------- | ---- | ---- | ---- | ----- |
>      | 18ms    | 64ms     | 5ms     | 5ms  | 3ms  | 4ms  | 99ms  |
>
>    - **Precision** **quantization**
>
>      The current implementation uses 32‑bit floating‑point arithmetic. Converting parts of the network to 16‑bit or even 8‑bit precision can further decrease memory bandwidth and accelerate inference without substantial accuracy degradation.
>
> Building on the optimization schemes outlined above, we applied targeted speed enhancements and achieved a throughput of 14.1 FPS, nearly 1.4 times faster than ESAM. At the same time our approach continues to surpass ESAM in overall performance.
>
> | Method              | AP↑  | AP₅₀↑ | AP₂₅↑ | FPS↑ |
> | ------------------- | ---- | ----- | ----- | ---- |
> | ESAM                | 43.4 | 65.4  | 80.9  | 10.6 |
> | Ours                | 46.2 | 67.9  | 81.7  | 10.1 |
> | Ours+fp16           | 44.8 | 66.3  | 81.2  | 13.6 |
> | Ours+voxel size*1.5 | 44.4 | 65.7  | 81.0  | 14.1 |
>
> **Q3: Threshold of Learning-based Mask Integration (LMI).**
>
> **A3:** In practice, we utilize a fixed threshold across **all** test scenarios, which consistently delivers robust results. To minimize false positives, we deliberately set this threshold at a conservatively high level. Consequently, minor perturbations around this value exert only negligible impacts on overall performance. To empirically substantiate these assertions, we conducted evaluations of our method using thresholds ranging from 0.5 to 0.9. As shown in the table below, the results exhibit virtually identical AP scores across this range, thereby confirming the robustness of our LMI strategy to threshold variations.
>
> | Threshold     | AP↑  | AP₅₀↑ | AP₂₅↑ |
> | ------------- | ---- | ----- | ----- |
> | without LMI   | 45.6 | 66.9  | 81.0  |
> | 0.5           | 46.0 | 67.6  | 81.5  |
> | 0.6           | 46.2 | 67.4  | 81.4  |
> | 0.8           | 46.1 | 67.6  | 81.8  |
> | 0.9           | 46.0 | 67.4  | 81.8  |
> | 0.7 (current) | 46.2 | 67.9  | 81.7  |
>
> **Q4: Differences between the proposed long-term memory mechanism and ESAM’s** **query** **merging.**
>
> **A4:** We highlight two essential differences between our method and query merging in ESAM, one during training and the other at inference.
>
> 1. **Training supervision：Direct tracking versus indirect feature alignment.** **Per-object precise supervision.** Our network learns with an explicit one-to-one matching loss. Each object instance is supervised to establish accurate correspondences across frames. **Set-based coarse supervision in ESAM.** Query merging compresses every instance into a single centroid vector. The loss then pulls centroids of the same object together while pushing different objects apart.
>
>    The feature alignment through centroids in query merging of ESAM can be problematic since: **a) Viewpoint and occlusion changes.** Consecutive LiDAR sweeps expose different surfaces of the same object. Forcing identical centroids penalizes legitimate variations. **b) Loss of spatial structure.** Averaging points erases intra-instance geometry, so the network cannot tell stable regions from naturally varying ones. **c) Over-regularization.** Aggressive centroid overlap reduces sensitivity to fine details, which harms accuracy in cluttered scenes.
>
> 2. **Inference behaviour: Lifecycle management.** In ESAM, once a query is created, it remains in memory forever, even if it was a false positive. Without any expiry, these spurious entries keep matching in later frames and propagate errors. We introduce a simple but effective policy: a track that stays unmatched for more than T frames is removed from memory. By discarding stale or erroneous objects, we curb cascading false matches and deliver steadier online tracking.
>
> Beyond the gaps above, ESAM relies on hand-crafted rules to fuse appearance and set-level features. Our model instead adopts a learnable fusion module, allowing the network itself to decide the contribution of each cue.
>
> **Q5: Is the threshold in Mask integration (LMI) robust to fragment granularity and could it** **learn this threshold adaptively?**
>
> **A5:** We thank the reviewer for this insightful question.
>
> 1. **Robustness to Fragment Granularity.**
>
>    The learned mask integration (LMI) threshold is not highly sensitive to fragment granularity. We use a single fixed threshold across datasets and categories without per‑scenario tuning, and observe consistent performance. This robustness comes from the fusion rule that combines geometric proximity with feature‑similarity scores, which reduces reliance on any one threshold. We further validate our method by sweeping the threshold and reporting per-class metrics over four representative categories, ordered from fine to coarse namely picture, chair, table, bed. Picture captures small planar instances that often break into many tiny fragments, chair contains multiple thin parts and yields medium to high fragmentation, table has a large contiguous surface with fewer thin elements, and bed is a bulky object that produces coarse fragments.  As shown in the table below, our method maintains strong results across different thresholds and fragment granularities, and it consistently outperforms the variant without LMI. These findings indicate that LMI is insensitive to granularity variations.
>
>    | Threshold     | picture↑ | chair↑ | table↑ | bed↑ |
>    | ------------- | -------- | ------ | ------ | ---- |
>    | without LMI   | 43.9     | 63.2   | 52.1   | 45.9 |
>    | 0.6           | 44.2     | 63.9   | 52.0   | 48.5 |
>    | 0.8           | 44.4     | 64.1   | 52.5   | 47.5 |
>    | 0.9           | 43.9     | 64.0   | 52.2   | 47.3 |
>    | 0.7 (current) | 44.4     | 63.6   | 52.9   | 48.4 |
>
> 2. **Adaptive Threshold Learning.**
>
>    We appreciate this insightful suggestion and plan to explore two complementary strategies in the future work:
>
>    - **Statistical Separation:** Fit the confidence scores of *matched* versus *unmatched* fragments to two Gaussian distributions, then choose the threshold that maximizes their separation (e.g., via the Bayesian decision boundary).
>
>    - **Network‑Predicted Thresholds:** Extend the model so that, in addition to mask outputs, it predicts a per‑object threshold at inference time. We would train these thresholds end‑to‑end by evaluating merge quality, allowing each instance to adapt its own threshold based on scene context.
>
>
> These future directions promise to make our mask‑fusion process even more flexible, while our current results already demonstrate strong robustness under fixed‑threshold operation. Thank the reviewers for this insightful idea!

---

> > ### Comment · Reviewer_Xyr3 · 2025-08-06
> >
> > Thank you for the detailed rebuttal. I appreciate the clarifications on the long-term memory mechanism and the robustness of the LMI threshold. Most of my concerns have been addressed, so I will maintain my original score. I look forward to the updated analyses in the revised paper.

---

> > > ### Author Response · Authors · 2025-08-07
> > > **Thank Reviewer Xyr3 for the positive recommendation**
> > >
> > > Thank you for maintaining a borderline accept recommendation. We will incorporate the additional results and technical details following your suggestions.

---

### Note · Authors · 2025-08-12

Dear ACs and Reviewers,

Thank you for your time reviewing our paper. We appreciate the positive advantages highlighted by all reviewers:

- Clear problem framing that recasts online 3D segmentation as instance tracking with identity-aware temporal reasoning.
- Effective design with LTM for long-range identity, STM for immediate temporal context, and spatial consistency learning to curb over-segmentation.
- Competitive accuracy with real-time throughput and thorough ablations, plus cross-dataset generalization.

During rebuttal, we address most reviewers' concerns.

- **Xyr3.** Requested runtime and memory profiling and asked about differences from ESAM and the robustness of the LMI threshold. We provided end-to-end latency and memory comparisons, clarified LTM versus ESAM query merging in both training and inference, and showed that a single fixed LMI threshold is robust across ranges and categories. **With most concerns addressed, the reviewer kept a borderline-accept recommendation.**
- **iEi7.** Asked for stronger evidence of real-time performance and generalization. We added a more detailed efficiency breakdown and reported results on Matterport3D that consistently outperform ESAM. **With most concerns addressed, the reviewer has maintained the initial positive score.**
- **jX3W.** Questioned whether gains come from temporal reasoning and asked for cross-dataset ablations. We reported per-module contributions, added ID-switch analysis to isolate temporal stability, and included ScanNet200 to SceneNN transfer ablations showing consistent gains. **Reviewer kept a borderline-accept recommendation.**
- **XxuF.** Raised concerns about writing quality, organization, similarities to ESAM, and edge deployment. We clarified preliminaries and motivation, contrasted our method with ESAM in supervision, memory lifecycle, and learnable fusion, and showed simple speedups such as precision reduction and voxel resizing that increase FPS while retaining accuracy. **The reviewer raised no additional questions.**

We thank the reviewers for the constructive guidance. We believe the concerns have been substantively addressed with new analyses and experiments, and the planned revisions will further improve clarity and completeness.

Best regards,

The authors of Paper 2965

---

### Decision · Program_Chairs · 2025-09-17

**Decision:**

Accept (poster)

**Comment:**

All reviewers recommend accepting this paper, congratulations!

The paper addresses online 3D segmentation by formulating it as an instance tracking task. It leveraged both long-term and short-term memory mechanisms and spatial consistency learning. This design yields strong performance on ScanNet, SceneNN, and related benchmarks, with thorough ablations and high FPS demonstrating promising application potential.

The main concerns raised were around architectural complexity and its derived consequence on deployment. Although a simpler method is more preferable, this does not prevent the work meeting the criteria for acceptance. As noted by the reviewers, please clarify some writing parts.

Overall, the contributions are meaningful and timely. I recommend acceptance, with revisions to improve clarity, efficiency discussion, and positioning relative to prior work.